# Effect of Centrality on Physical Activity in Late Childhood: A 1-Year Prospective Cohort Study

**DOI:** 10.3390/children11091084

**Published:** 2024-09-04

**Authors:** Kazuya Tamura, Takashi Saito, Yuya Ueda, Ryo Goto, Naoki Yamada, Toshihiro Akisue, Rei Ono

**Affiliations:** 1Department of Rehabilitation Science, Kobe University Graduate School of Health Sciences, Kobe 654-0142, Japan; tam.finswim@gmail.com (K.T.); yuyaueda@port.kobe-u.ac.jp (Y.U.); akisue@med.kobe-u.ac.jp (T.A.); 2Department of Rehabilitation, Tokushima University Hospital, Tokushima 770-8503, Japan; t.saito@tokushima-u.ac.jp; 3Division of Rehabilitation Medicine, Kobe University Hospital, Kobe 650-0047, Japan; rgoto@med.kobe-u.ac.jp; 4Department of Rehabilitation, Hyogo Cancer Center, Akashi 673-0021, Japan; mr.ymc1023@gmail.com; 5Department of Physical Activity Research, National Institutes of Biomedical Innovation, Health and Nutrition, Settsu 566-0002, Japan; 6Department of Public Health, Kobe University Graduate School of Health Sciences, Kobe 654-0142, Japan

**Keywords:** physical activity, social network analysis, friendship network, centrality, school age

## Abstract

Background/Objectives: Engaging in physical activity (PA) is crucial for children’s physical and mental health, with PA in childhood influencing lifelong activity levels. However, PA during childhood tends to decrease with age. Childhood friendship networks influence various health behaviors, including physical activity. Centralities are objective measures of an individual’s position and role in friendship networks. The relationship between centrality and PA is inconsistent. This study aimed to determine how centrality affects changes in PA in late childhood longitudinally and to investigate the distribution of centrality in the network. Methods: This prospective cohort study recruited fourth- and fifth-grade children (9–11 years old). A total of 143 children participated. We calculated three centralities—in-degree, closeness, and betweenness—based on social network analysis (SNA). PA was assessed using the physical activity questionnaire for older children (PAQ-C). To explore the relationship between centralities and the percentage change in PA, a multivariate logistic regression analysis was performed. Results: Children with higher closeness had a significantly higher rate of decrease in PA after adjusting for confounding factors. There was no significant association between betweenness and percentage change in PA (*p* = 0.66) or in-degree and percentage change in PA (*p* = 0.21). Conclusions: This study highlights the importance of considering social network dynamics in PA interventions, particularly for children with high social closeness. Future research should incorporate objective PA measures and explore broader social networks to enhance intervention strategies, especially for Generation Z and Alpha, who experience unique opportunities and motivations for PA due to pervasive digital environments.

## 1. Introduction

Engaging in physical activity (PA) has positive effects on both physical and mental health [1]. PA in childhood influences PA in adulthood and beyond [2]. However, PA during childhood tends to decrease with age [3]. This is because children who spend more screen time (an indicator of sedentary behavior) than that recommended by guidelines are increasing annually [4]. More screen time and less PA are independently associated with metabolic risk in children [5]. Therefore, addressing the decrease in PA during childhood is crucial for long-term health.

Friendship networks formed during childhood significantly impact various health behaviors, including PA [6,7]. As networks are formed, individual positions within the network become well-defined. Centralities are objective measures of an individual’s position and role in friendship networks [8]. People with high centrality are considered to have a central position in a network. The representative indicators of centrality are in-degree, closeness, and betweenness [9]. Centrality is positively associated with health behaviors [10]. However, negative correlations have also been reported, such as in adolescents with high closeness centrality who tend to have drinking habits [11]. These mixed findings highlight the need for further investigation into how centrality relates to PA.

Existing research on the relationship between centrality and PA has produced inconsistent results. A positive association between in-degree centrality and organized PA has been reported [12]. Meanwhile, there were negative associations between in-degree centrality and participation in group sports among both sexes and between closeness centrality and participation in group sports among girls [13]. Therefore, there is insufficient evidence of a relationship between centrality and PA. Moreover, the cross-sectional design of previous research restricts the ability to establish causal relationships. This study aims to address these gaps by longitudinally investigating how centrality influences changes in PA during late childhood, thereby contributing valuable insights into this complex relationship. In addition to exploring the relationship between centrality and PA, this study also examines the distribution of centrality measures within the network. By providing an overview of the distribution of in-degree, closeness, and betweenness centrality, this research aims to offer insights into the structural characteristics of the network. This additional analysis helps contextualize the centrality measures and provides a more comprehensive understanding of the network dynamics.

## 2. Materials and Methods

### 2.1. Study Population and Design

This study was a prospective cohort study. In October 2020, we enlisted fourth- and fifth-grade students (ages 9–11) from two public elementary schools in Kobe, Japan. The study had a follow-up duration of one year. Participants included children who completed the self-reported questionnaire. Children with missing data, those attending special-needs classes, and those who changed schools during follow-up were excluded. We excluded children attending special-needs classes because they may not attend classes in the same classroom as other children. They were not exposed to the same conditions as other children to form friend networks. We provided an explanation of the study protocol to the children, as well as to the principals and teachers, and secured informed consent from both the children and their parents. The study received approval from the Institutional Review Board of the Kobe University Graduate School of Health Sciences (approval number 545-5).

### 2.2. Data Collection

All data were collected using self-report questionnaires and anthropometric measurements. Self-report questionnaires were administered during homeroom periods at each class and school. For students who missed the homeroom period or made errors in their responses, the questionnaires were completed after class time with support from graduate student measurers. This approach ensured that all students had the opportunity to provide accurate responses, and any issues with the questionnaire completion were addressed promptly. Anthropometric measurements were based on data collected in 2020, conducted at the schools. Demographic characteristics, PA, centrality, screen time at baseline (October 2020), and PA at follow-up (October 2021) were used. Anthropometric measurements were conducted by elementary school teachers and staff during the first semester of 2020. Examples of specific questions are provided in the section for each item.

### 2.3. Measures

#### 2.3.1. Physical Activity

PA was assessed using the physical activity questionnaire for older children (PAQ-C), which demonstrates adequate internal consistency and test-retest reliability (Cronbach’s α = 0.80; intraclass correlation coefficient = 0.83) [14]. The PAQ-C includes nine items that can be computed, rated on a 1–5 Likert scale, where higher scores reflect greater levels of physical activity. For example, we asked, “How many times in the last seven days have you done any of the exercises listed in the items below?” and asked the respondents to check “none”, “1–2 times”, “3–4 times”, “5–6 times”, or “7 or more times” for 22 items, including playing tag, baseball, swimming, soccer, etc. Individual PAQ-C scores were calculated as the mean scores of the nine items, and they ranged from 1.00 (lowest activity level) to 5.00 (highest activity level). The percentage change in PA was calculated using the following formula:(PAQ-C score at follow-up − PAQ-C score at baseline)/PAQ-C score at baseline × 100.

#### 2.3.2. Social Network Analysis

Social network analysis (SNA) is a specialized analytical method used to chart and examine the relationships and interactions among individuals [15]. In this study, we considered school-grade networks to represent the entire network. We then used relationships within school-grade networks to analyze the centrality of the individual within the network [16]. We conducted SNA using the following procedure: first, we provided each child with a list of classmates, including their names and attendance numbers. Second, we obtained the friends lists from each child by asking them the following: “Please list the best friends who you play with or talk to in your grade.” Children were permitted to name up to 10 of their closest friends [16]. Third, using the igraph package in R, we mapped the children as nodes and connected them to each other with lines based on friends lists [17]. This mapping of nodes and lines is the social network data and is the output for each grade level.

#### 2.3.3. Centrality

Centralities were calculated based on social networks mapped through the SNA. The adjacency matrix created during the SNA process was then converted into a graph object. Centrality measures were extracted from this graph object. The specific centrality measures calculated included in-degree, closeness, and betweenness. The explanations of these centrality measures and the R scripts used for their calculation are provided below [9]:

In-degree: The number of nominations an individual has received [13]. Children with a high in-degree level usually have influence or prestige [18].

Closeness: A measure of how close an individual is to other network members. It is calculated as the inverse of the sum of the lengths of the shortest path between an individual and all the other children in the entire network [19]. Children with high closeness can spread information the fastest way [13].

Betweenness: Measures how often an individual appears on the shortest paths between other children in the entire network [20]. Children with high betweenness connect with other individuals, facilitate communication between other children, and play the role of bridge [9].

The R scripts used for these calculations are as follows:# Example R scripts for calculating centrality measures# Load the igraph package
library (igraph)
# Create a graph object from the adjacency matrix
graph <- simplify (graph_from_adjacency_matrix (adj_matrix, mode = “directed”), remove.loops = TRUE)
# Calculate centrality measures
In-degree <- degree (graph, mode = “in”, loops = F, normalized = T)

Closeness <- closeness (graph, mode = “all”, weights = NULL, normalized = T)

Betweenness <- betweenness (graph, directed = T, weights = NULL, normalized = T)


The raw value of centralities depends on the number of people (n) in the social network and, therefore, takes different ranges of values for each network. The maximum value of each centrality was {n − 1} for in-degree, {1/(n − 1)} for closeness, and {(n − 1) (n − 2)/2} for betweenness. Centralities take values ranging from 0 to 1 by dividing the raw values of centralities by the maximum values [21]. Through this normalization procedure, the centrality values can be compared among different friend networks.

#### 2.3.4. Confounding Factors

Data on age, sex, school grade, and screen time were gathered through a self-reported questionnaire. Height (in cm) and weight (in kg) were recorded through anthropometric measurements conducted at the elementary schools. Body mass index (BMI) was computed from the participants’ height and weight (kg/m^2^). For screen time, participants reported their average daily duration of cellular phone use and video game play. For example, in the case of cellular phones, the question is “Approximately how many hours a day do you usually spend on your cellular phone texting, calling, or playing games?”. The children respond from a list of nine of the following: “never”, “about 30 min”, “about 1 h”, “2 h”, “3 h”, “4 h”, “5 h”, “6 h”, and “at least 7 h” [22]. In this study, screen time using cellular phones and playing video games were categorized into “≤ 2 h” and “> 2 h.” [23]

### 2.4. Analysis

Data are reported as counts (percentages) for categorical variables and as mean ± standard deviation for continuous variables. Univariate and multivariate regression analyses were conducted to examine the relationship between each baseline centrality and the percentage change in PA. In all analyses, the explanatory variable was centrality at baseline, and the objective variable was the percentage change in PA. Each of the three centralities at the baseline was entered into a separate regression equation. The following variables were used as confounding factors in the multivariate regression analyses: sex, school grade, BMI, cellular phone screen time, and video games screen time [14,23]. The standardized partial regression coefficient (β), 95% confidence intervals, *p* values, and coefficient of determination (R^2^) were computed. Statistical significance was defined as *p* < 0.05. Data analysis was conducted using the free software R (version 4.0.3; R Foundation for Statistical Computing, Vienna, Austria; https://www.R-project.org) [24].

## 3. Results

Out of 192 children initially recruited, 27 were excluded due to missing data or attendance in special-needs classes at baseline, and 22 were excluded for dropping out during follow-up. Consequently, 143 children were included in the final analysis.

The flowchart of the study is presented in Figure 1.

Table 1 presents the characteristics of all the participants.

Figure 2 illustrates the distribution of the three centrality measures. These values indicate:

Closeness (mean = 0.47): A high average value suggests that the network is relatively efficient, with students being well-connected through short paths.

In-Degree (mean = 0.16): A moderate average value indicates that students have a relatively limited number of direct connections, with no single student having an exceptionally high number of direct connections.

Betweenness (mean = 0.035): A low average value shows that students play a minimal role as intermediaries, implying a relatively direct flow of information within the network.

Table 2 displays the results from both univariate and multivariate regression analyses. The univariate regression analysis showed that children with higher closeness experienced a significantly greater decrease in PA (*p* < 0.01). There was no significant association between betweenness and percentage change in PA (*p* = 0.66) or in-degree and percentage change in PA (*p* = 0.21). After adjusting for sex, school grade, BMI, cellular phone screen time, and video game screen time, higher closeness predicted a significantly higher rate of decrease in PA (*p* = 0.01). In the multivariate analyses, there was no significant association between betweenness and percentage change in PA (*p* = 0.77) or between in-degree and percentage change in PA (*p* = 0.42), as in the univariate analyses.

Figure 3 shows scatter plots and regression lines to illustrate the association between the three centralities and percentage change in physical activity.

## 4. Discussion

This study investigated the association between the three centralities and the change in PA longitudinally. Overall, PA in children tended to decrease throughout the year. Regarding the relationship between centrality and rate of change in PA, higher closeness was associated with a greater decrease in PA over 1 year. In-degree and betweenness were not significantly associated with the rate of change in PA.

The results indicate that the network of elementary school students within the same grade exhibits specific characteristics in terms of centrality measures. The high average value of closeness centrality (mean = 0.47) demonstrates that the network is efficient, allowing for the rapid dissemination of information and facilitating effective communication among students [21,25]. The moderate average value of in-degree centrality (mean = 0.16) reveals that while students are connected to some peers, the connections are relatively limited, with no student having a significantly higher number of direct links [26,27]. This suggests a relatively balanced network in terms of direct connectivity. The low average value of betweenness centrality (mean = 0.035) indicates that the network lacks significant intermediaries. Information flows directly between students with fewer bottlenecks or gatekeepers. This structure may contribute to the network’s efficiency but also suggests a potential lack of central figures who mediate communication [21,25]. These findings offer insights into the network’s communication dynamics, suggesting an efficient network with relatively direct information flow and balanced connectivity. Understanding these characteristics can help in designing interventions or strategies to enhance communication and interactions within similar educational settings.

This study indicates that centrality may influence PA after 1 year. Higher closeness is associated with a greater decrease in PA. A cross-sectional study reported a negative association between participation in group sports and closeness in adolescent girls [13]. The present study supports the study of Marqués-Sánchez et al. and adds a new finding that closeness has a longitudinal negative effect on PA [13]. In-degree and betweenness were not associated with changes in PA. While numerous prior studies have explored the connection between in-degree centrality and PA, the findings have been inconsistent. One study found a positive link between in-degree and participation in organized PA among adolescent boys, whereas another study indicated that adolescents involved in group sports generally exhibit lower in-degree centrality [12,13]. These studies were cross-sectional and did not measure changes in PA. The present study suggests that in-degree may be transversely associated with PA, whereas it may not be associated with changes in PA. Betweenness and PA have been investigated in adolescents, but no significant associations have been found [28]. The longitudinal results in this study support the cross-sectional results of the study of Sawka et al., and betweenness may not have an impact on PA. Further investigations are required to confirm these associations [28].

In this study, closeness was a factor that predicted a decrease in PA. Intervention studies have demonstrated that individuals with high closeness play an important role in the spreading of healthy behaviors [29,30]. A simulation intervention study targeting children with high closeness to improve PA showed increasing PA in the population [29]. In contrast, another study that conducted a similar intervention in the real world did not observe any effect of intervention for closeness on PA [30]. Closeness has been used as a centrality that plays a positive role in health behaviors; however, it may have a negative effect on PA in the real world.

This study had two strengths. First, we investigated changes in PA longitudinally. Using a longitudinal design, we investigated the relationship between centrality and PA. Second, this study investigated three centralities, whereas most previous studies investigated only in-degree centrality. However, this study has three limitations. First, centrality indices are highly dependent on the characteristics and structure of the network, which can vary significantly across different networks. This variability makes it challenging to compare our findings directly with those of previous studies. Specifically, the average values and distributions of centrality measures can differ markedly based on network density, node roles, and overall structure. Therefore, the results from previous research may not always be applicable to our study’s context. This limitation underscores the need for caution when generalizing our findings to other networks or settings. Second, we used a subjective scale to measure the PA. Nevertheless, the Japanese version of the PAQ-C used in this study is a questionnaire with verified reliability and validity. Future studies should perform both objective PA assessment using accelerometers and subjective PA assessment using questionnaires. Third, the measurements were conducted over a limited area. Future studies are required to investigate this in various regions. Finally, one of the key limitations of this study is the relatively low R-squared value of 0.05 obtained from the regression analysis. While the low R-squared value indicates a limited explanatory power, it highlights the need for further research to identify and include additional variables or to explore alternative methodological approaches to better understand the factors influencing the outcome.

This study provides important information for novel interventions to improve PA in children. Previous studies have reported the effectiveness of interventions such as an after-school activity program and smartphone games in improving PA among children during childhood [31,32]. In addition to these intervention methods, attention should be paid to intervention targets, as children with high closeness are at risk of decreased PA in future research. Additionally, Generation Z and Generation Alpha, who have grown up with constant access to the internet and smartphones, may have distinct opportunities and motivations for physical activities, including play [33,34]. This evolving landscape suggests a need to reassess how these opportunities and motivations are addressed in interventions. Furthermore, their social networks are likely to extend beyond traditional school settings, highlighting the necessity for research that adopts a broader perspective to fully understand their social connections and their impact on PA.

## 5. Conclusions

This study shows that higher social closeness is linked to a greater decrease in physical activity (PA) over 1 year, suggesting that strong social ties might contribute to reduced PA. In-degree and betweenness had no significant impact. These findings highlight the need for more research.

Future research should use objective PA measures and explore broader social networks, especially for Generation Z and Alpha, to improve PA interventions.

## Figures and Tables

**Figure 1 children-11-01084-f001:**
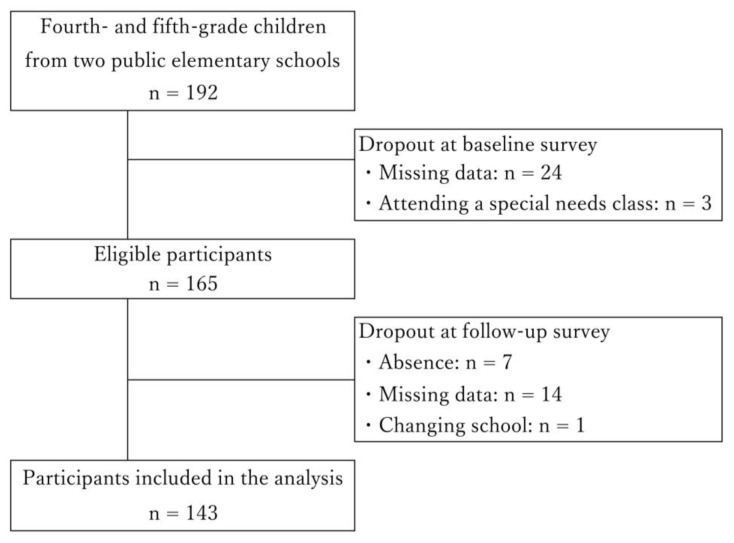
Flowchart of this study.

**Figure 2 children-11-01084-f002:**
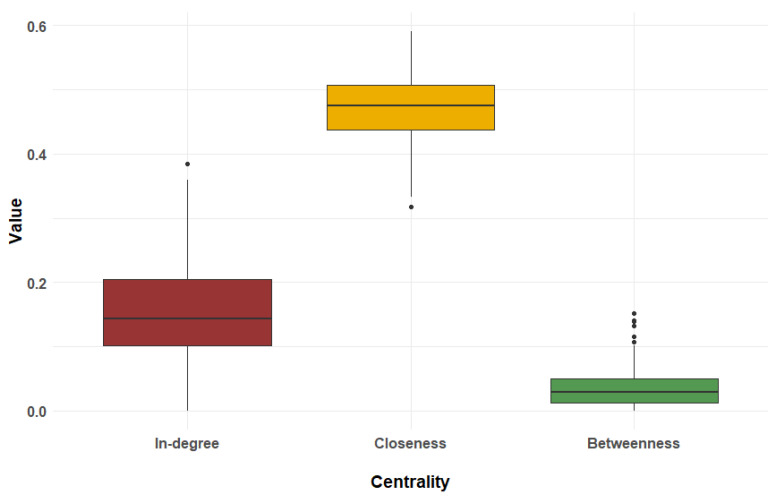
Distribution of centrality measures by type: in-degree, closeness, and betweenness.

**Figure 3 children-11-01084-f003:**
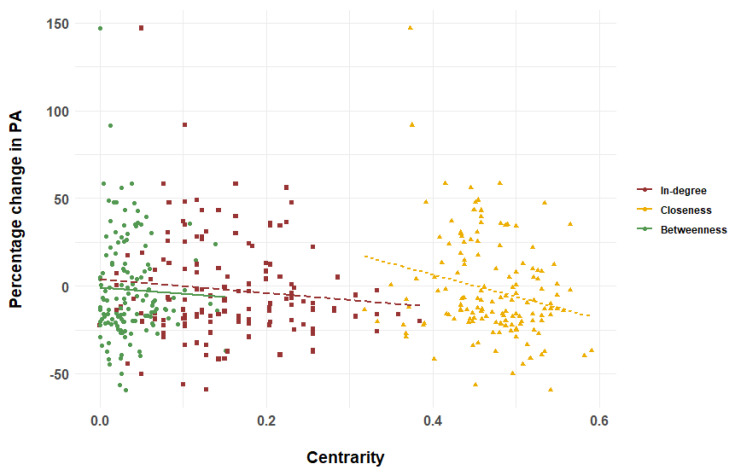
Relationship between centrality and percentage change in physical activity (PA).

**Table 1 children-11-01084-t001:** Characteristics of the participants (n = 143).

Variables	Values
Age (years)	10.0 ± 0.7
Sex (female)	68 (48%)
Grade (fourth grade)	75 (52%)
Body mass index (kg/m^2^)	17.3 ± 2.7
Screen time > 2 h/day	
Cellular phone	15 (10%)
Video game	25 (17%)
Centrality	
In-degree	0.16 ± 0.08
Closeness	0.47 ± 0.05
Betweenness	0.035 ± 0.03
Physical activity	
Baseline	2.9 ± 0.8
Follow-up	2.8 ± 0.8
Percentage change in physical activity (%)	−2.3 ± 29.1

Data are presented as mean ± standard deviation or n (%). Mean: the arithmetic average of a set of values, calculated by dividing the sum of all values by the number of values; standard deviation: a measure of the amount of variation or dispersion in a set of values; it indicates how much the values typically deviate from the mean.

**Table 2 children-11-01084-t002:** Univariate and multivariate regression analyses assessing the relation between centrality and percentage change in physical activity.

	Univariate				Multivariate			
	β	95% CI	*p*	R^2^	β	95% CI	*p*	R^2^
In-degree	−0.10	−0.27, 0.06	0.21	<0.01	−0.07	−0.24, 0.11	0.42	<0.01
Closeness	−0.23 *	−0.39, −0.07	<0.01	0.05	−0.22 *	−0.40, −0.05	0.01	0.04
Betweenness	−0.04	−0.20, 0.13	0.66	<0.01	−0.03	−0.20, 0.15	0.77	<0.01

* *p* < 0.01. β, standardized partial regression coefficient—A measure of the strength and direction of the relationship between a predictor variable and the outcome variable, expressed in standard deviation units. CI, confidence interval—A range of values that is likely to contain the true value of the parameter being measured with a certain level of confidence. *p*, *p*-value—A measure of the strength of the evidence against the null hypothesis. R^2^, adjusted R^2^—A statistical measure that represents the proportion of the variance for a dependent variable that is explained by independent variables in a regression model, adjusted for the number of predictors used in the model. All multivariate regression analyses were adjusted for school grade, sex, body mass index, cellular phone screen time, and video game screen time.

## Data Availability

The data presented in this study are available on request from the corresponding author. The data are not publicly available due to specific restrictions imposed by the ethics committee, which do not permit data sharing due to privacy concerns and confidentiality requirements.

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
