# Peer review of "Effect of Centrality on Physical Activity in Late Childhood: A 1-Year Prospective Cohort Study"

_children, 2024, doi:10.3390/children11091084_

Round 1

Reviewer 1 Report

Comments and Suggestions for Authors

Dear Authors,

I have recently had the opportunity to read your article titled “Effect of Centrality on Physical Activity in Late Childhood: A 1-Year Prospective Cohort Study”. I found it to be quite interesting, especially considering the limited number of studies available on this topic.

Overall, the article is well-written and clear in its results and conclusions. However, there are certain issues that need to be addressed:

1) Please, revise the Style in cite 19 (page 3).

2) Is the self-report quesionnaire for screen time a valid and reliable questionnaire for children? If yes, please provide more information about it.

3) I would like to observe the correlation coefficients between confounding factors and PA. Moreover, I would like to see the effect size and power of your sample and results.

4) It is necessary to improve the quality of Figure 1, it is not readable.

5) Please provide every Unit in Table 1, and separate outcomes in packs for a better understanding.

6) Although the Discussion is correct, it is very brief. Please try to discuss your findings in more depth in comparison to previous studies (even if they are few) and, above all, the implications that your results have for children's health.

7) What are the recommendations that you can make, based on your results and experience, to increase PA in children? What strategies can be implemented in educational centers? I think it is very relevant that this is reflected in your manuscript, it is the IMPORTANCE of your study.

Author Response

Thank you for your kind comments. Please see the attachment.

Reviewer 2 Report

Comments and Suggestions for Authors

This study investigated the association between the three centralities and the change in PA longitudinally. Overall, PA in children tended to decrease throughout the year. Regarding the relationship between centrality and rate of change in PA, higher closeness was associated with a greater decrease in PA over 1 year. In-degree and betweenness were not significantly associated with the rate of change in PA.

Please try to show a few tipes of questions to see if are aproppiate with children age and level of understanding.

The statistic table is correct but the explenations are poor.

Comments on the Quality of English Language

I just find a few character mistake. Must read and correct again the manuscript

Author Response

(The authors gave the same response as above.)

Reviewer 3 Report

Comments and Suggestions for Authors

I think this study will not contribute to the literature.

I think the subject is not original.

The discussion and conclusion part is not enough. The findings section is also not enough.

The introduction could not fully reflect the study. 

The English language is insufficient in some parts.

Comments on the Quality of English Language

I think this study will not contribute to the literature. I think the subject is not original. The discussion and conclusion part is not enough. The findings section is also not enough. The introduction could not fully reflect the study.  The English language is insufficient in some parts. In short, the study does not seem useful. 

Author Response

(The authors gave the same response as above.)

Round 2

Reviewer 2 Report

Comments and Suggestions for Authors

The authors took into account the observations and acted accordingly

Reviewer 3 Report

Comments and Suggestions for Authors

There is no difference between my previous decision and my current decision. I think this issue will not attract the attention of the authors. I think it's not original.

Comments on the Quality of English Language

There are still some bugs in the English language.